# Determination of the Cutoff Frequency of Smoothing Filters for Center of Pressure (COP) Data via Kinetic Energy in Standing Dogs

**DOI:** 10.3390/s25185843

**Published:** 2025-09-18

**Authors:** Julia Wegscheider, Christiane Lutonsky, Nadja Affenzeller, Masoud Aghapour, Barbara Bockstahler, Christian Peham

**Affiliations:** 1Section of Physical Therapy and Rehabilitation, Small Animals Surgery, Clinical Centre for Small Animal Health and Research, Clinical Department for Small Animals and Horses, University of Veterinary Medicine, 1210 Vienna, Austria; julia.wegscheider@vetmeduni.ac.at (J.W.); christiane.lutonsky@vetmeduni.ac.at (C.L.); nadja.affenzeller@vetmeduni.ac.at (N.A.); masoud.aghapour@vetmeduni.ac.at (M.A.); barbara.bockstahler@vetmeduni.ac.at (B.B.); 2Behavioral Medicine, Clinical Centre for Small Animal Health and Research, Clinical Department for Small Animals and Horses, University of Veterinary Medicine, 1210 Vienna, Austria; 3Movement Science Group, Clinical Centre for Equine Health and Research, Clinical Department for Small Animals and Horses, University of Veterinary Medicine, 1210 Vienna, Austria

**Keywords:** cutoff frequency, kinetic energy, low pass, smoothing data, filtering data

## Abstract

**Highlights:**

This article highlights the theoretical basis, practical implementation, and potential benefits of determining the cutoff frequency of smoothing filters for motion data by imposing an upper limit on kinetic energy. To demonstrate the feasibility of this approach, we used postural sway in standing dogs, represented by their COP (center of pressure) movement. This universal approach is not tied to any specific species or movement.

**What are the main findings?**
An upper limit of the kinetic energy can be directly used to define the cutoff frequency for the low-pass filter to smooth motion data.Our biologically motivated approach leads to a similar recommendation for the cutoff frequency for stationary dogs as the engineering-based approach but is easy to define because of the energy limit.

**What is the implication of the main finding?**
Smoothing filters are necessary to reduce noise, eliminate outliers, and prepare the data for further analysis or visualization.A critical factor in applying such filters is the choice of cutoff frequency, which directly influences the quality and interpretability of the filtered data.

**Abstract:**

This article investigates the determination of the optimal cutoff frequency of smoothing filters for motion data based on kinetic energy. The underlying hypothesis is that an upper limit of kinetic energy can serve as a basis for setting the cutoff frequency. To illustrate this, postural sway in standing dogs was analyzed based on the movement of their center of pressure (COP). The method was tested on 12 clinically healthy dogs that met specific inclusion criteria. The results show that a cutoff frequency of 6 Hz, derived from an individual kinetic energy calculated from the COP velocity of 5 cm/s, provides the best representation of postural sway, while 10 Hz filtered data was sufficient in only 6 of 12 dogs, and unfiltered data unsuitable in 12 of 12 dogs. The study highlights that the choice of cutoff frequency is crucial for data quality and proposes a biologically motivated method based on kinetic energy. This method could lead to more precise and meaningful results in motion analysis.

## 1. Introduction

Motion data analysis plays a central role in numerous scientific and technical disciplines, including biomechanics, robotics and sports science. Reliable and meaningful results, however, crucially depend on the quality of the underlying data. Smoothing filters are used to reduce noise, eliminate outliers, and prepare the data for further analysis or visualization. A critical factor when applying such filters is the choice of cutoff frequency, which directly influences the quality and interpretability of the filtered data [1,2,3].

The cutoff frequency of a low-pass filter determines which frequency components of the motion data are retained and which are filtered out as noise. Choosing a cutoff frequency is a balancing act: if it is too low, important motion details can be suppressed, while if it is too high, noise and unwanted interference will be present in the data. Beyond improving data quality, choosing the correct cutoff frequency offers other advantages, such as the removal of outliers and erroneous data points, improving the structuring and organization of the data, and enabling a clearer and more meaningful visualization [1].

Conventionally, the cutoff frequency is determined based on the subject’s movement speed or the expected frequency spectrum of the signal [2]. However, these methods are not without limitations. Studies have shown that the choice of cutoff frequency can significantly influence the results of motion analysis [3], yet the rationale behind this choice is often underreported or absent.

In research focusing on the center of pressure (COP), filtering techniques are widely applied, but their implementation is rarely discussed in detail. Most studies simply refer to previous research. In human, canine, and equine research, low-pass filters, such as Butterworth [4,5,6,7,8,9] or Chebyshev types [10,11] are frequently applied. While a cutoff frequency of 10 Hz is typical in human studies [3,4,6,7,10,12], alternative values such as 25 Hz [13], 15 Hz [11], and 12.5 Hz [14] have also been reported.

Considering the widespread use of filtering techniques in human research, it is surprising that these methods have not yet been consistently adopted in veterinary research. Most canine and equine studies do not report the application of a filtering method. However, a few equine studies have implemented a fourth-order Butterworth filter with a cutoff frequency of 15 Hz [8,9]. In dog research using the COP trajectory, two publications explicitly describe the use of a low-pass filter with cutoff frequencies of 5 Hz [15] and 10 Hz [16].

The locomotion of living beings is highly cost-efficient, meaning that the energy expenditure of each task or movement is kept as low as possible and thus limited [17].

This article highlights the theoretical basis, practical implementation, and potential benefits of determining the cutoff frequency of smoothing filters for motion data by imposing an upper limit on kinetic energy. As far as we know, this is a procedure. The aim of this article is to present a method for determining the optimal cutoff frequency based on the kinetic energy of the motion. The underlying hypothesis is that there is an upper limit to the kinetic energy that can serve as a basis for determining the cutoff frequency of smoothing algorithms. To demonstrate the feasibility of this approach, we used postural sway in standing dogs, represented by their COP movement [16]. Given the established role of dogs as a model organism for human aging [18], their use demonstrates a transferable methodological approach that is not limited to a specific species or movement.

## 2. Materials and Methods

### 2.1. Ethics Statement

This study was approved by the Ethics and Animal Welfare Committee of the University of Veterinary Medicine, Vienna in accordance with the University’s guidelines for Good Scientific Practice guidelines and national legislation (ETK-148/10/2021). In addition, the owners’ consent to this investigation was part of the approval.

### 2.2. Animals and Inclusion Criteria

This study included 12 client-owned dogs. The inclusion criteria were an age <50% of their fractional lifespan [19,20], had no orthopedic, neurological, or visual diseases, and had a minimum body mass of 10 kg. All dogs underwent a general clinical examination, including visual gait assessment, orthopedic and neurologic examination, and objective gait analysis using a pressure measurement plate to exclude lame dogs (FDM Type 2, Zebris Medical GmbH, Allgäu, Germany) [16,20].

The dogs consisted of four Labrador retrievers, two Border collies, one standard poodle, one flat-coated retriever, one Malinois, one Irish terrier, one pointer, and one mixed breed dog. The mean age and body mass were 3.39 ± 1.79 years and 22.68 ± 5.43 kg, respectively; see Table 1.

### 2.3. Equipment and Measurement Procedure

All dogs were assessed during quiet standing on a pressure measurement plate (FDM Type 2, Zebris Medical GmbH, Allgäu, Germany) equipped with 15,360 sensors covering an area of 203 × 54.2 cm, with a sampling frequency of 100 Hz. Each sensor measured 0.72 × 0.72 cm. To ensure a secure footing, the plate was covered with a 1 mm-thick, non-slip black rubber mat. For quality control, all measurement procedures were video recorded using a Panasonic NV-MX500 camera (Panasonic, Kadoma, Osaka, Japan), as previously described [21].

### 2.4. Objective Gait Analysis

All dogs were walked over the pressure plate until a minimum of five valid passes per paw were recorded. A pass was considered valid if the dog crossed the plate in a straight line without speed changes, head turns, or leash tension. Gait symmetry was confirmed by a symmetry index of peak vertical force and vertical impulse below 3%. The walking speed across the plate had to remain within ±0.3 m/s, with an acceleration not exceeding ±0.5 m/s^2^ [22,23].

### 2.5. Static Posturography

For this study only the data of the pressure mat were used. After a short break, static posturography was performed during quiet standing. This was defined as a stance with all limbs perpendicular to the plate without any body, head, tail, limb, or paw movements. For this, the owner stood in front of the animal to maintain its attention during the measurement procedure. After each measurement, the animal was rewarded with a treat and asked to rest. The required measurement duration was set to 10 s of quiet standing and was repeated three times [20].

### 2.6. Data Analysis

The fractional lifespan (FLS) was calculated by the following formula:FLS = 13.620 + (0.0276 × body height in cm) − (0.1186 × body mass in kg)

The data was analyzed using a custom software Pressure Analyzer (Michael Schwanda, version 4.9.3.0), which was then exported to Microsoft Excel 2016.

The kinetic energy of a dog swaying at a speed of 5 cm/s was assumed as the upper limit [16,21,22]. This means that the kinetic energy of the standing dog calculated from the COP coordinates should be below this limit (see Table 1). For a time series of 10 s with a sample frequency of 100 Hz, no more than 50 data points should be above the limit (*p* < 0.05). The kinetic energy was calculated according to Formula (2).$$E_{kin}\;=\;\frac{mv^{2}}{2}\;whereas\;m\ldots .mass\;of\;the\;dog,\;v\ldots speed\;of\;the\;COP\;(5\frac{cm}{s}\;for\;the\;limit)$$(1)$$v\;=\;\frac{ds}{dt}\;=\;\dot{s},s=\sqrt{x^{2}\;+\;y^{2}}$$
where x and y are the coordinates and s is the position vector of the COP.

The COP coordinates were unfiltered and band-limited with a Butterworth (4th order) low-pass filter with a cutoff frequency of 6 (Wn = Fc/(Fs/2) = 0.12) and 10 Hz (Wn = Fc/(Fs/2) = 0.2) (see Figure 1). For this purpose, we used the “butter” function of the MATLAB software (version 24b) to create the filter characteristics and the “filtfilt” function to smooth the data. We used the “filtfilt” function to avoid phase shifts. The kinetic energy of each dog at a swaying speed of 5 cm/s was calculated and set as the upper limit. The number of kinetic energy values below the upper limit was determined. If 95% of all values were below the maximum energy (upper limit), the cutoff frequency was considered sufficient.

### 2.7. Statistics

Counts were averaged for each dog from three trials (10 s each). The data were grouped into unfiltered and filtered data with a cutoff frequency of 6 Hz and a cutoff frequency of 10 Hz, respectively. The data were checked for normal distribution with a Kolmogorov–Smirnov test. The groups were compared using a repeated-measures ANOVA (general linear model) and a Bonferroni post hoc test using SPSS version 29.01.0.

## 3. Results

Our results show that a cutoff frequency of 10 Hz significantly improves the data representation, while a cutoff frequency of 6 Hz is sufficient to meet our criterion that only 5% of the time series violate our upper bound. The criteria were met by 10 of 12 dogs at 6 Hz, by only 6 of 12 dogs at 10 Hz, and by no dog with unfiltered COP data. The differences between 6 and 10 Hz are clearly illustrated by the kinetic energy plot with the calculated limit (Figure 2). This is not as evident in the temporal progression of the signal (Figure 1). Unfiltered data, on the other hand, violates our criterion, does not provide sufficient representation, and may lead to incorrect analyses and interpretations.

## 4. Discussion

The strengths and weaknesses of the pressure mat and the data acquisition protocol are discussed in detail in [16,21,22].

The 3 dB cutoff frequency is the frequency at which the output power of a filter drops to half of its maximum passband power [24]. Typically, the cutoff frequency is determined based on the subject’s expected frequency spectrum of the signal [2].

However, this is more complex in standing mammals, as no movement is clearly visible.

Deriving suitable filtering and smoothing methods requires a detailed knowledge of biological systems and their dynamics. Therefore, individually tailored methods should be applied [25].

We know that standing mammals, including dogs, constantly sway their bodies to stabilize their posture [7,26]. This sway is influenced by a variety of factors. According to [7], these variables can be divided into four groups: position, dynamics, frequency, and stochastic variables.

While [2] proposes a frequency-based approach, we believe that a dynamic approach based on kinetic energy, which considers the animal’s mass in addition to speed, is simpler and more meaningful. Mass plays a crucial role in an animal’s achievable speed [27]. It is an important factor used by experienced canine experts (such as veterinarians or trainers) to estimate a dog’s potential movement speed. This is especially true when the center of gravity (CG) of a standing dog, roughly represented by the measured COP on a force plate, is stabilized [28].

Therefore, we propose defining the individual upper limit for standing dogs in our cohort as a COP speed of 5 cm/s. For a 25 kg dog, this results in a kinetic energy of 0.03125 Nm. This can be interpreted as a dog’s mass (relative to the acceleration due to gravity of approximately 9.81 m/s^2^) being lifted by approximately 0.13 mm. This value provides a plausible upper energetic limit for involuntary swaying while standing.

The criteria were met by 10 of 12 dogs at 6 Hz, by only 6 of 12 dogs at 10 Hz, and by no dog with unfiltered COP data. Therefore, we conclude that unfiltered data provide an unsuitable representation of postural sway, while choosing a cutoff frequency of 10 Hz is sufficient for 50% of the data. The best representation of postural sway is found at a cutoff frequency of 6 Hz, while for two dogs, either the energy limit was too low or the cutoff frequency was too high (Table 2 and Figure 3).

While a cutoff frequency of 10 Hz is typical in human studies [3,4,6,7,10,12], higher cutoff frequencies such as 25 Hz [13], 15 Hz [11], and 12.5 Hz [10] have also been reported. According to this study, the used cutoff frequencies are too high for dogs. This may be because humans have a higher body mass and thus higher kinetic energy at the same oscillation speed, leading to higher cutoff frequencies. This is consistent with studies in horses (with higher body mass) where a higher cutoff frequency of 15 Hz [8,9] was used, while studies in dogs used lower frequencies of 5 Hz [15] and 10 Hz [16]. If we had selected dogs with a higher mass (more than 50 kg), we might have obtained similar results. Selecting the cutoff is often challenging because it must be within a specific range and can also be too low. The 3 dB cutoff frequency is the frequency at which the output power of a filter drops to half its maximum pass power [24]. Therefore, we consider a reduction in the cutoff frequency to be critical, as it could violate the Nyquist–Shannon theorem criterion that determines the lower limit.

Another explanation for the outliers (e.g., dog 11 and dog 12) could be that the dogs exhibited higher energy levels due to hidden movements such as panting and tail-wagging.

It is known that synchronization between respiration and postural sway has been observed in humans [29]. Our experience suggests that both panting and tail-wagging have a significant influence on postural sway. For this reason, we attempted to exclude dogs or repeated the measurements if the dogs were panting or tail-wagging during recording to standardize measurements. Excessive kinetic energy can be used to detect hidden movements and exclude these animals/data from investigations.

### Limitations of the Methods and the Role of Frequency

However, these methods are not without limitations. Determining maximum kinetic energy is much easier for moving dogs (e.g., dogs that are walking or running) than for stationary animals. Calorimetric studies would be necessary to determine the exact energy expenditure of a stationary dog [30]. However, since only an upper limit is needed to determine the cutoff frequency of the low-pass filter, we consider a rough estimate based on the sway velocity and known body mass to be sufficient.

Our observations are consistent with previous studies showing that the average sway frequency in newborn foals is low (<1 Hz) and decreases further with age [31]. The influence of age was minimized by the selection of dogs within a certain range of age. The mean age were 3.39 ± 1.79 years. Furthermore, ref. [31] supports our suggestion to use a cutoff frequency of 6 Hz for standing dogs. Considering the Nyquist-Shannon sampling theorem, a bandwidth of more than 2 Hz is required to adequately capture the relevant/wanted signals.

## 5. Conclusions

Our biologically motivated approach leads to a similar recommendation for the cutoff frequency for stationary dogs as the engineering-based approach of [2], namely a cutoff frequency of 6 Hz. We believe this concept can be easily extended to moving animals. Determination is easier because running speed (walking, trotting, or galloping) can be used directly to define the upper limit of kinetic energy and thus to determine the cutoff frequency for the low-pass filter. We are convinced that this method of denoising the data will lead to better and more meaningful results.

## Figures and Tables

**Figure 1 sensors-25-05843-f001:**
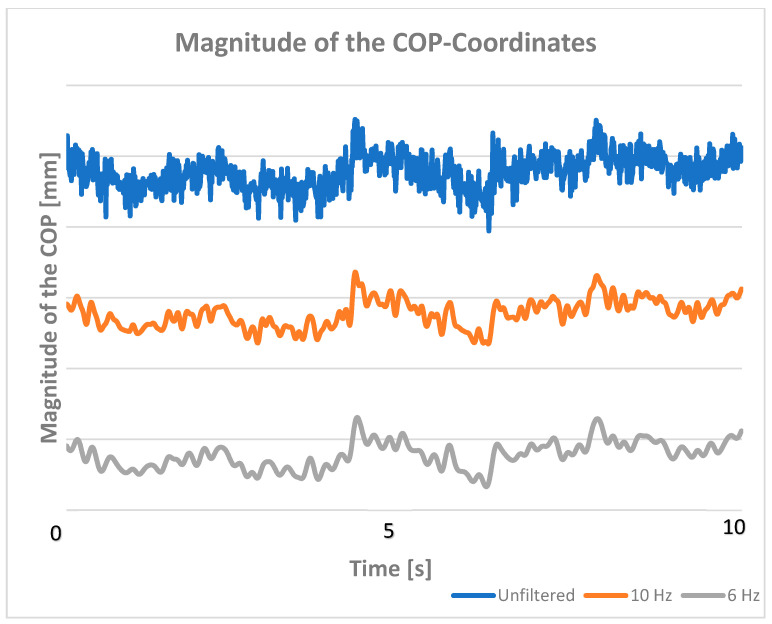
The magnitude of the position vector of the COP unfiltered (blue), and after low-pass filtering at 10 Hz (orange) and at 6 Hz (gray).

**Figure 2 sensors-25-05843-f002:**
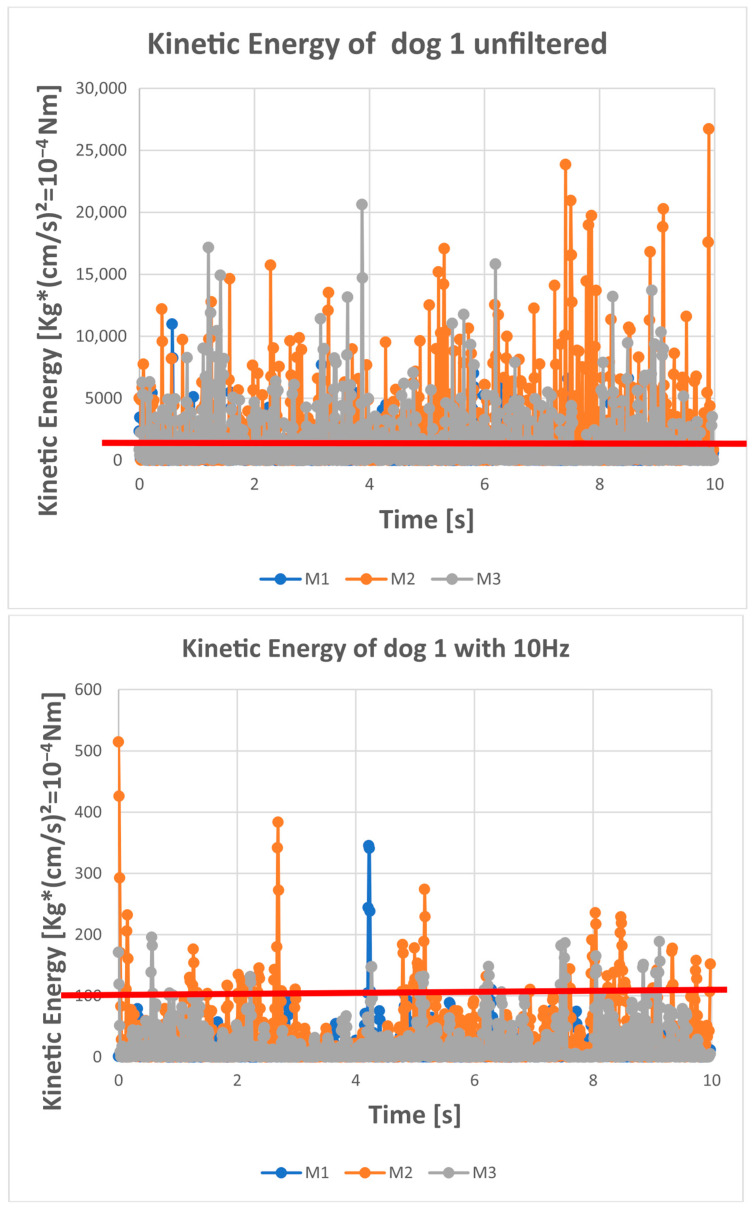
The kinetic energy of the COP over time for each trial (each measurement) of dog 1. M1 is the first 10 s measurement (quiet standing trial). M2 is the second 10 s measurement (quiet standing trial). M3 is the third 10 s measurement (quiet standing trial). The red line in the graph is the upper limit of the kinetic energy (= no value should be above this limit). For this dog, the upper limit was determined by a speed of 5 cm/s and the dog had a mass of 13.2 kg (=165 × 10^−4^ Nm). It is obvious that unfiltered data exceed the upper limit very often (regularly); this is reduced by a 10 Hz low-pass filter and occurs very rarely for the 6 Hz filtered data.

**Figure 3 sensors-25-05843-f003:**
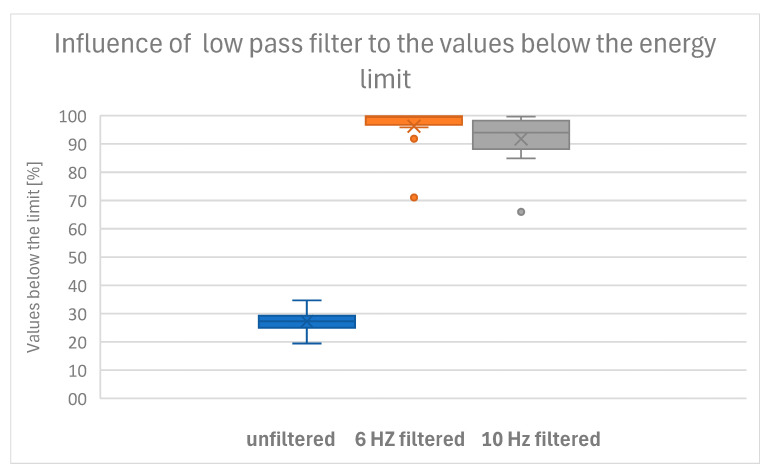
Boxplot of the percentages below the threshold shows that only a low-pass filter of 6 Hz meets our criterion that 95% of the kinetic energy values are under the upper limit.

**Table 1 sensors-25-05843-t001:** List of the dogs (f—female; m—male; fc—female castrated) and their individual upper limit of kinetic energy.

Dog	Breed	Sex	Age [Years]	Mass [Kg]	Limit of Kinetic Energy [Kg × cm^2^/s^2^ = 10^−4^ Nm]
1	Mixed-breed	fc	2.7	13.2	165.00
2	Standard Poodle	m	6.0	26.0	325.00
3	Border Collie	m	5.9	24.0	300.00
4	Flat-Coated Retriever	m	1.1	32.5	406.25
5	Labrador	f	2.1	20.7	258.75
6	Border Collie	f	2.0	16.0	200.00
7	Malinois	f	3.9	20.6	257.50
8	Irish Setter	m	5.9	21.0	262.50
9	Labrador	f	1.3	20.9	261.25
10	Labrador	f	3.5	21.3	266.25
11	Pointer	fc	2.2	26.0	325.00
12	Labrador	m	4.1	30.0	375.00

**Table 2 sensors-25-05843-t002:** The kinetic energy **V**alues **b**elow the upper **L**imit (**VBL**). If more than 95% (950 out of 1000) are below the limit, the filtering is considered sufficient. The green numbers (>95) indicate the percentage of values below the limit and indicate sufficient low-pass filtering. The red numbers (<95) indicate the percentage of VBL and indicate insufficient low-pass filtering.

Dogs	VBL	VBL6	VBL10	VBL [%]	VBL6 [%]	VBL10 [%]
**1**	** 284 **	** 996 **	** 986 **	** 28.4 **	** 99.6 **	** 98.6 **
**2**	** 280 **	** 999 **	** 985 **	** 28.0 **	** 99.9 **	** 98.5 **
**3**	** 270 **	** 993 **	** 924 **	** 27.0 **	** 99.3 **	** 92.4 **
**4**	** 276 **	** 959 **	** 868 **	** 27.6 **	** 95.9 **	** 86.8 **
**5**	** 267 **	** 999 **	** 963 **	** 26.7 **	** 99.9 **	** 96.3 **
**6**	** 194 **	** 993 **	** 929 **	** 19.4 **	** 99.3 **	** 92.9 **
**7**	** 265 **	** 994 **	** 951 **	** 26.5 **	** 99.4 **	** 95.1 **
**8**	** 347 **	** 999 **	** 997 **	** 34.7 **	** 99.9 **	** 99.7 **
**9**	** 242 **	** 997 **	** 974 **	** 24.2 **	** 99.7 **	** 97.4 **
**10**	** 245 **	** 995 **	** 930 **	** 24.5 **	** 99.5 **	** 93.0 **
**11**	** 295 **	** 919 **	** 849 **	** 29.5 **	** 91.9 **	** 84.9 **
**12**	** 304 **	** 711 **	** 660 **	** 30.4 **	** 71.1 **	** 66.0 **

All values differ significantly, *p* < 0.001 (tested with a general linear model—ANOVA for repeated measurements (Appendix A).

## Data Availability

The original contributions presented in this study are included in the article. Further inquiries can be directed to the corresponding author.

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
