# Peer review of "Determination of the Cutoff Frequency of Smoothing Filters for Center of Pressure (COP) Data via Kinetic Energy in Standing Dogs"

_sensors, 2025, doi:10.3390/s25185843_

Round 1
Reviewer 1 Report
Comments and Suggestions for Authors
This study focuses on a narrow yet important aspect concerning the determination of the optimal cutoff frequency of smoothing filters for motion data based on kinetic energy. Although the topic and main findings may appear intuitive within the field of human-induced vibration, it is well established that most human motions occur within a low-frequency band (e.g., walking: 1.5 Hz to 2.5 Hz; jumping: 2.5 Hz to 3.5 Hz; bouncing: 2.0 Hz to 3.5 Hz), a principle that also applies to animal movement. Consequently, the novelty of this study appears limited. Nevertheless, the paper is generally well organized and presents a clear motivation. However, major revisions are required to more effectively communicate the study's contributions, and the figures should be improved for better clarity and presentation.
Detailed comments are listed as follows:
1.It is well established that most human motions occur within a low-frequency band, a principle that also applies to animal movement. Consequently, the novelty of this study appears limited.
- Why choose dog as the research subject? Are the findings applicable to other animals?
- Why use the center of pressure instead of displacement of CENTER OF MASS to investigate the postural sway?
- The cutoff frequency of 6 Hz is objective. From Fig. 1, the 10 Hz cutoff frequency has the similar filtering effect.
- All the figures are not well presented in a unified manner.
- The Discussion section is meaningful. However, measurement protocolis also important in high quality data collection. The authors are encouraged to include these aspects in this section.
Author Response
Many thanks to the reviewers for their valuable contributions to this article.
Rev1
This study focuses on a narrow yet important aspect concerning the determination of the optimal cutoff frequency of smoothing filters for motion data based on kinetic energy. Although the topic and main findings may appear intuitive within the field of human-induced vibration, it is well established that most human motions occur within a low-frequency band (e.g., walking: 1.5 Hz to 2.5 Hz; jumping: 2.5 Hz to 3.5 Hz; bouncing: 2.0 Hz to 3.5 Hz), a principle that also applies to animal movement. Consequently, the novelty of this study appears limited. Nevertheless, the paper is generally well organized and presents a clear motivation. However, major revisions are required to more effectively communicate the study's contributions, and the figures should be improved for better clarity and presentation.
Detailed comments are listed as follows:
1.It is well established that most human motions occur within a low-frequency band, a principle that also applies to animal movement. Consequently, the novelty of this study appears limited.
Reply: You're right that there are established methods. However, we believe that the kinetic energy approach is biologically and physically driven, as all species have limited energy available and use it very cost-efficiently. As we know, this approach is unique.
We added the sentence (L86): As far as we know, is this procedure new.
- Why choose dog as the research subject? Are the findings applicable to other animals?
Reply: This is a very interesting question. At the University of Veterinary Medicine Vienna, we use pressure mats in the clinical routine in small animals for lameness evaluation and objective control of therapy. Furthermore dogs are a well established model for aging (see: Mazzatenta A, Carluccio A, Robbe D, Giulio CD, Cellerino A. The companion dog as a unique translational model for aging. Semin Cell Dev Biol. 2017 Oct;70:141-153. doi: 10.1016/j.semcdb.2017.08.024.)
This article is part of the results of the project funded by the Austrian Science Fund with the title: “Aging and CDS in dogs influence postural control of dogs“.
We replaced the sentence: “This universal approach is not tied to any specific species or movement.” by :”Since dogs are a well-established model for human aging [18], we believe that by using dogs we illustrate a universal approach that is not tied to a specific species or movement.”
[18] is the reference Mazzaatena et al. 2017.
3.Why use the center of pressure instead of displacement of CENTER OF MASS to investigate the postural sway?
Reply: You're right that the movement of the center of mass perfectly describes postural sway. However, we used a pressure mat, and the movement of the center of pressure is a projection of the center of mass/gravity in the measurement plane (the movement of the center of pressure is used to control the position of the center of mass/gravity).
We have added a supporting reference [16] to the sentence: “To demonstrate the feasibility of this approach, we used postural sway in standing dogs, represented by their COP movement [16].”
4.The cutoff frequency of 6 Hz is objective. From Fig. 1, the 10 Hz cutoff frequency has the similar filtering effect.
Reply: Yes, your absolutely right, that the differences between the signal filtered with 6 Hz and 10 Hz aren't obvious in Fig. 1. But the differences are very clearly visible in Fig. 2, where the kinetic energy is plotted.
We added (L177-L179): The differences between 6 and 10 Hz are clearly illustrated by the kinetic energy plot with the calculated limit (Fig. 2). This is not as evident in the temporal progression of the signal (Fig. 1).
5. All the figures are not well presented in a unified manner.
Reply: The y-axis is now unified labeled as: Kinetic Energy [Kg*(cm/s)²=10-4Nm]
6. The Discussion section is meaningful. However, measurement protocolis also important in high quality data collection. The authors are encouraged to include these aspects in this section.
Reply: We added to the discussion section: The strengths and weaknesses of the pressure mat and the data acquisition protocol are discussed in detail in [16, 21, 22].
Reviewer 2 Report
Comments and Suggestions for Authors
The following logical progression in the introduction is understandable.
----------------
1) Smoothing filters are used to reduce noise, eliminate outliers, and prepare the data for further analysis or visualization. A critical factor in applying such filters is the choice of cutoff frequency, which directly influences the quality and interpretability of the filtered data.
2) Studies have shown that the choice of cutoff frequency can significantly influence the results of motion analysis [3], yet the rationale behind this choice is often underreported or absent.
3) The aim of this article is to present a method for determining the optimal cutoff frequency based on the kinetic energy of the motion.
----------------
First of all, why are you using dog data for verification? There is a lack of explanation of the purpose of the research and its scientific significance. It is unclear whether you chose dogs to verify your proposed method or whether you are proposing a method for analyzing dog behavior.
Author Response
Many thanks to the reviewers for their valuable contributions to this article.
Rev2
The following logical progression in the introduction is understandable.
----------------
1) Smoothing filters are used to reduce noise, eliminate outliers, and prepare the data for further analysis or visualization. A critical factor in applying such filters is the choice of cutoff frequency, which directly influences the quality and interpretability of the filtered data.
2) Studies have shown that the choice of cutoff frequency can significantly influence the results of motion analysis [3], yet the rationale behind this choice is often underreported or absent.
3) The aim of this article is to present a method for determining the optimal cutoff frequency based on the kinetic energy of the motion.
Reply: You're right that smoothing filters are necessary and there are established methods. However, we believe that the kinetic energy approach is biologically and physically driven, as all species have limited energy available and use it very cost-efficiently. As we know, this approach is unique.
We extended the statement: This article highlights the theoretical basis, practical implementation, and potential benefits of determining the cutoff frequency of smoothing filters for motion data by imposing an upper limit on kinetic energy. As far as we know, is this procedure new.
----------------
First of all, why are you using dog data for verification? There is a lack of explanation of the purpose of the research and its scientific significance. It is unclear whether you chose dogs to verify your proposed method or whether you are proposing a method for analyzing dog behavior.
Reply: This is a very interesting question. At the University of Veterinary Medicine Vienna, we use pressure mats in the clinical routine in small animals for lameness evaluation and objective control of therapy. Furthermore dogs are a well established model for aging (see: Mazzatenta A, Carluccio A, Robbe D, Giulio CD, Cellerino A. The companion dog as a unique translational model for aging. Semin Cell Dev Biol. 2017 Oct;70:141-153. doi: 10.1016/j.semcdb.2017.08.024.)
This article is part of the results of the project funded by the Austrian Science Fund with the title: “Aging and CDS in dogs influence postural control of dogs“.
We replaced the sentence: “This universal approach is not tied to any specific species or movement.” by :”Since dogs are a well-established model for human aging [18], we believe that by using dogs we illustrate a universal approach that is not tied to a specific species or movement.”
[18] is the reference Mazzaatena et al. 2017.
Reviewer 3 Report
Comments and Suggestions for Authors
This manuscript presents a biologically motivated method to determine the cutoff frequency of smoothing filters for center-of-pressure (COP) data in dogs, based on an upper kinetic energy limit. The study is well-structured, the methodology is clearly described, and the results are supported by appropriate statistical analyses. The approach is innovative and of interest to both veterinary biomechanics and motion analysis communities. The manuscript is overall of good quality and merits publication in Sensors. However, I recommend minor revision to improve clarity, strengthen justification, and ensure reproducibility.
Major comments
The assumption that 5 cm/s represents a reasonable upper limit of COP sway speed is central to the method, but its rationale is not fully explained. Please provide additional justification (e.g., prior literature, pilot observations, or physiological reasoning) to support this choice.
While the results indicate significance (p < 0.001), the manuscript would benefit from reporting the full ANOVA statistics (F-value, degrees of freedom, and effect size, e.g., partial η²). This will improve transparency and reproducibility.
Since kinetic energy is directly proportional to body mass, please expand on whether larger dogs tended to show systematically higher or different cutoff values. This would strengthen the biological plausibility of the method.
Minor comments
Some sentences are lengthy and repetitive (e.g., repeated phrasing of “theoretical basis, practical implementation, and potential benefits”). Please simplify to improve readability.
Current figure legends are quite minimal. Please make them more self-contained by explaining the meaning of lines, thresholds, and the interpretation of results (e.g., explicitly state that the red line = upper kinetic energy limit).
In the methods section, please provide the exact filter parameters (order, normalized cutoff frequency) used in MATLAB. This will ensure reproducibility.
In Table 1, the kinetic energy unit is presented as [10^-4 Nm]. Consider clarifying this in the caption or text to avoid confusion for readers unfamiliar with the scaling.
The discussion of panting/tail wagging as potential confounders is valuable. It would be helpful to briefly mention how future work could systematically address or correct for such hidden movements.
Author Response
Many thanks to the reviewers for their valuable contributions to this article.
This manuscript presents a biologically motivated method to determine the cutoff frequency of smoothing filters for center-of-pressure (COP) data in dogs, based on an upper kinetic energy limit. The study is well-structured, the methodology is clearly described, and the results are supported by appropriate statistical analyses. The approach is innovative and of interest to both veterinary biomechanics and motion analysis communities. The manuscript is overall of good quality and merits publication in Sensors. However, I recommend minor revision to improve clarity, strengthen justification, and ensure reproducibility.
Major comments
The assumption that 5 cm/s represents a reasonable upper limit of COP sway speed is central to the method, but its rationale is not fully explained. Please provide additional justification (e.g., prior literature, pilot observations, or physiological reasoning) to support this choice.
We have added the references on which this speed limit is based in the methods section: “The kinetic energy of a dog swaying at a speed of 5 cm/s was assumed as the upper limit [16,21,22].”
While the results indicate significance (p < 0.001), the manuscript would benefit from reporting the full ANOVA statistics (F-value, degrees of freedom, and effect size, e.g., partial η²). This will improve transparency and reproducibility.
We will add the following table from SPSS to supplement:
Paarweise Vergleiche |
||||||
Maß: MASS_1 |
||||||
(I) Filterung |
(J) Filterung |
Mittelwertdifferenz (I-J) |
Std.-Fehler |
Sig.b |
95% Konfidenzintervall für Differenzb |
|
Untergrenze |
Obergrenze |
|||||
1 |
2 |
-725,833* |
37,638 |
<,001 |
-831,975 |
-619,692 |
3 |
-629,500* |
33,502 |
<,001 |
-723,976 |
-535,024 |
|
2 |
1 |
725,833* |
37,638 |
<,001 |
619,692 |
831,975 |
3 |
96,333* |
14,164 |
<,001 |
56,391 |
136,276 |
|
3 |
1 |
629,500* |
33,502 |
<,001 |
535,024 |
723,976 |
2 |
-96,333* |
14,164 |
<,001 |
-136,276 |
-56,391 |
|
Basiert auf geschätzten Randmitteln |
||||||
*. Die Mittelwertdifferenz ist in Stufe ,05 signifikant. |
||||||
b. Anpassung für Mehrfachvergleiche: Bonferroni. |
Since kinetic energy is directly proportional to body mass, please expand on whether larger dogs tended to show systematically higher or different cutoff values. This would strengthen the biological plausibility of the method.
You are right, the mass and limit values ​​are shown in Table 1.
Furthermore, we discussed this fact:
“While [2] proposes a frequency-based approach, we believe that a dynamic approach based on kinetic energy, which considers the animal's mass in addition to speed, is simpler and more meaningful. Mass plays a crucial role in an animal's achievable speed [27].”
and
“While a cutoff frequency of 10 Hz is typical in human studies [3,4,6,7,10,12], higher cutoff frequencies such as 25 Hz [13], 15 Hz [11], and 12.5 Hz [10] have also been reported. According to this study the used cutoff frequencies are too high for dogs. This may be because humans have a higher body mass and thus higher kinetic energy at the same oscillation speed, leading to higher cutoff frequencies. This is consistent with studies in horses (with higher body mass) where a higher cutoff frequency of 15 Hz [8,9] was used, while studies in dogs used lower frequencies of 5 Hz [15] and 10 Hz [16].”
Minor comments
Some sentences are lengthy and repetitive (e.g., repeated phrasing of “theoretical basis, practical implementation, and potential benefits”). Please simplify to improve readability.
Current figure legends are quite minimal. Please make them more self-contained by explaining the meaning of lines, thresholds, and the interpretation of results (e.g., explicitly state that the red line = upper kinetic energy limit).
We changed the caption of figure1 in: “The magnitude of the position vector of the COP unfiltered (blue), and after low-pass filtering at 10 Hz (orange) and at 6 Hz (grey).”
Caption of Figure 2 and 3 and table 2 was changed accordingly: “Figures 2: The kinetic energy of the COP over time for each trial (each measurement) of dog 1…”
“Figure 3: Boxplot of the percentages below…”
“Table 2: The kinetic energy Values below the upper Limit (VBL)….”
In the methods section, please provide the exact filter parameters (order, normalized cutoff frequency) used in MATLAB. This will ensure reproducibility.
We added the normalized cut-off to this sentence: “The COP-coordinates were unfiltered and band-limited with a Butterworth (4th order) low-pass filter with a cutoff frequency of 6 (Wn=Fc/(Fs/2)=0.12) and 10 Hz (Wn=Fc/(Fs/2)=0.2) (See Fig. 1).”
In Table 1, the kinetic energy unit is presented as [10^-4 Nm]. Consider clarifying this in the caption or text to avoid confusion for readers unfamiliar with the scaling.
We added to clarify: [Kg*cm²/s²=10-4Nm]
The discussion of panting/tail wagging as potential confounders is valuable. It would be helpful to briefly mention how future work could systematically address or correct for such hidden movements.
We added to the discussion: “Excessive kinetic energy can be used to detect hidden movements and exclude these animals/data from investigations.”
Reviewer 4 Report
Comments and Suggestions for Authors
Dear,
Please find my comments attached.
Kind regards

Author Response
Many thanks to the reviewers for their valuable contributions to this article.
Rev4
This study analyzes postural sway in dogs, a parameter often linked to neurological, musculoskeletal, or vestibular disorders. Establishing reliable filtering methods ensures that assessments of postural control in veterinary medicine are accurate. By refining the data quality in center of pressure (COP) measurements, the method can help detect subtle balance impairments that might otherwise be overlooked, enabling earlier diagnosis or better monitoring of conditions affecting mobility. The article emphasizes that an inappropriate cutoff frequency produces misleading results. This means veterinarians and researchers can avoid misinterpretations that could lead to inaccurate treatment planning or ineffective rehabilitation protocols.
The topic of the paper is interesting, and the study has potential for publication; however, it also contains numerous shortcomings, outlined below:
It is necessary to improve the abstract. In the Methods section of the abstract, additional information is needed regarding the frequencies used in the study, while the Results section would benefit from being expressed more quantitatively.
Reply: We added in the abstract: The results show that a cutoff frequency of 6 Hz, derived from an individual kinetic energy calculated from the COP velocity of 5 cm/s, provides the best representation of postural sway, while 10 Hz filtered data were sufficient in only 6 of 12 dogs, and unfiltered data unsuitable in 12 of 12 dogs.
In the Introduction chapter, the sentence at lines 54–55 lacks a proper reference to support the authors’ claim.
Reply: References were added: [1,2,3].
In Chapter 2 (Materials and Methods), several important details are missing:
Written informed consent from the dog owners regarding their animals’ participation in the study.
The study was approved by the ethics commission of the University of Veterinary Medicine Vienna (ETK-148/10/2021). The owner consent to this investigation was included in this approval.
- Specification of whether the kinematic measurements were conducted under 2D or 3D conditions, and if 2D, in which plane (frontal, sagittal, or horizontal).
Only pressure data were collected (FDM Type 2, Zebris Medical GmbH, Allgäu, Germany). The cameras were used for quality control to ensure that the dogs were standing. No kinematic data were measured.
- Clarification on whether the dogs underwent familiarization with the tensiometric platform prior to the objective gait analysis and static posturography.
Yes, the dogs had time to acclimate to the environment. The procedure is described in detail in [16, 21,22].
We added to the discussion section: The strengths and weaknesses of the pressure mat and the data acquisition protocol are discussed in detail in [16, 21, 22].
- A dedicated subchapter listing the variables of interest and their units of measurement.
In Figure 1, the term „figures“ representing lines should be written as the magnitudes rather than the plural „figures“. Additionally, it is necessary to specify which line color corresponds to which measurement condition and magnitude so that the Figure 1 is fully self-explanatory.
Reply: We changed the caption of figure1 in: “The magnitude of the position vector of the COP unfiltered (blue), and after low-pass filtering at 10 Hz (orange) and at 6 Hz (grey).”
The same comments apply to Figure 2, which also lacks a proper title, as well as to Table 2 and Figure 3.
Reply: Caption of Figure 2 and 3 and table 2 was changed accordingly: “Figures 2: The kinetic energy of the COP over time for each trial (each measurement) of dog 1…”
“Figure 3: Boxplot of the percentages below…”
“Table 2: The kinetic energy Values below the upper Limit (VBL)….”
The results in Table 2 should be presented consistently— either with or without boldface—using black font only, accompanied by a clear legend, rather than a mix of black and green.
Reply: We think that the combination of red (insufficient) and green (sufficient) numbers makes the result easy to understand for the reader.
Another issue concerns the range of dog body mass (13.2 kg to 30 kg), which may represent a limitation of the study. If so, this fact should be explicitly stated as part of the study’s limitations.
Reply: The inclusion of the dog's mass in the kinetic energy method (=mass*v²/2) does not limit the investigation.
We stated in the discussion section: “While a cutoff frequency of 10 Hz is typical in human studies [3,4,6,7,10,12], higher cutoff frequencies such as 25 Hz [13], 15 Hz [11], and 12.5 Hz [10] have also been reported. According to this study the used cutoff frequencies are too high for dogs. This may be because humans have a higher body mass and thus higher kinetic energy at the same oscillation speed, leading to higher cutoff frequencies. This is consistent with studies in horses (with higher body mass) where a higher cutoff frequency of 15 Hz [8,9] was used, while studies in dogs used lower frequencies of 5 Hz [15] and 10 Hz [16].”
Finally, in the Conclusion chapter, the text refers to running speed (walking, trotting, galloping), even though this parameter was not part of the study design, nor was it included in the figures, tables, or discussion. This creates confusion for the reader. The Conclusion chapter should instead be clearly and precisely structured to answer the study’s aims and hypotheses, which is currently not the case. In summary, the paper is confusing, lacks clarity in key sections, and remains incomplete in its current form.
Reply: As mentioned in the article, the concept of limited kinetic energy is a holistic approach that is not restricted to a specific task or movement as well as species.
For clarification, we add: We believe this concept can be easily extended to moving animals. Determination is easier because running speed (walking, trotting, galloping) can be used directly to define the upper limit of kinetic energy and thus to determine the cutoff frequency for the low-pass filter. We are convinced that this method of denoising the data will lead to better and more meaningful results.

Round 2
Reviewer 2 Report
Comments and Suggestions for Authors
To prove the hypothesis that the upper limit of kinetic energy can serve as a criterion for setting a cutoff frequency, we analyzed the postural sway of 12 clinically healthy dogs in a standing position based on center of pressure (COP) movement.
_The explanations in P3 #93-#95 have helped me understand the overall direction.
I_s the expression in P2 #45 correct?
"While 10 Hz filtered data were sufficient in only 6 of 12 dogs, unfiltered data was unsuitable in 12 of 12 dogs."
12 of 12 -> 6 of 12?
_Table 1
Mass from 13.2 to 32.5... Size is small to medium size.
I believe that including data on sizes between 40kg and 50kg would be more useful in demonstrating the generality of the hypothesis. This is closer to human size.
_Dog 11 and Dog 12 do not meet all the criteria. Can you explain why?
Table 1 shows that the dogs(11 and 12) appear to have a relatively high limit of kinetic energy.
Author Response
To prove the hypothesis that the upper limit of kinetic energy can serve as a criterion for setting a cutoff frequency, we analyzed the postural sway of 12 clinically healthy dogs in a standing position based on center of pressure (COP) movement.
_The explanations in P3 #93-#95 have helped me understand the overall direction.
I_s the expression in P2 #45 correct?
"While 10 Hz filtered data were sufficient in only 6 of 12 dogs, unfiltered data was unsuitable in 12 of 12 dogs."
12 of 12 -> 6 of 12?
Yes, this statement (10 Hz filtered data were sufficient in only 6 of 12 dogs, and unfiltered data unsuitable in 12 of 12 dogs.) is correct.
This can be seen in Table 2: All unfiltered values ​​are red, meaning that all 12 dogs failed our criterion, while with 10 Hz filtered data, the criterion was met for 6 of 12 dogs.
_Table 1
Mass from 13.2 to 32.5... Size is small to medium size.
I believe that including data on sizes between 40kg and 50kg would be more useful in demonstrating the generality of the hypothesis. This is closer to human size.
You are right that larger dogs are closer to humans, but unfortunately, we were unable to recruit larger dogs for this study.
We added to the discussion section (line 254-255): If we had selected dogs with a higher mass (more than 50 kg), we might have obtained similar results.
_Dog 11 and Dog 12 do not meet all the criteria. Can you explain why?
We added to the discussion section: .. (e. g. dog 11 and dog 12)… to explain this outliers.
Another explanation for the outliers (e. g. dog 11 and dog 12) could be that the dogs exhibited higher energy levels due to hidden movements such as panting and tail wagging.
It is known that synchronization between respiration and postural sway has been observed in humans [29]. Our experience suggests that both panting and tail wagging have a significant influence on postural sway. For this reason, we attempted to exclude dogs or repeated the measurements if the dogs were panting or tail wagging during recording to standardize measurements. Excessive kinetic energy can be used to detect hidden movements and exclude these animals/data from investigations.
Table 1 shows that the dogs(11 and 12) appear to have a relatively high limit of kinetic energy.
Yes, this is due to their masses of 26 and 30 kg. Dog 4 had the highest mass at 32.5 kg and a limit of 406.25 10-4 Nm. Nevertheless, dogs 2-4 are in a similar range and meet our criteria.
Reviewer 4 Report
Comments and Suggestions for Authors
Dear,
In the Introduction chapter, the way the authors formulated the study hypothesis is not well aligned with the stated research aims. This needs to be reformulated.
Regarding Chapter 2, Materials and Methods, the authors need to explicitly add in the text that the owners' consent of this investigation was included in the approval mentioned, i.e., the approval of the University of Veterinary Medicine. The manuscript currently lacks this information, which represents a shortcoming.
In addition, the text should also include an explanation related to the kinematic measurements. As the authors have clarified, the cameras were only used for quality control purposes to ensure that the dogs remained in a standing position. Therefore, the manuscript should clearly state that no kinematic variables were measured and that only data collected from the tensiometric platform were analyzed.
Kind regards
Author Response
In the Introduction chapter, the way the authors formulated the study hypothesis is not well aligned with the stated research aims. This needs to be reformulated.
We changed the statement in (L93-96): Given the established role of dogs as a model organism for human aging [18], their use demonstrates a transferable methodological approach that is not limited to a specific species or movement.
Regarding Chapter 2, Materials and Methods, the authors need to explicitly add in the text that the owners' consent of this investigation was included in the approval mentioned, i.e., the approval of the University of Veterinary Medicine. The manuscript currently lacks this information, which represents a shortcoming.
We added to the method section (L 104-105): In addition, the owners' consent to this investigation was part of the approval.
In addition, the text should also include an explanation related to the kinematic measurements. As the authors have clarified, the cameras were only used for quality control purposes to ensure that the dogs remained in a standing position. Therefore, the manuscript should clearly state that no kinematic variables were measured and that only data collected from the tensiometric platform were analyzed.
We added (L122): For quality control, all measurement procedures were video recorded using a Panasonic NV-MX500 camera (Panasonic, Kadoma, Osaka, Japan), as previously described [21].
And (L135): For this study only the data of the pressure mat were used.